# Two-qubit logic and teleportation with mobile spin qubits in silicon

Y. Matsumoto[1,3], M. De Smet[1,3], L. Tryputen[2], S. L. de Snoo[1], S. V. Amitonov[2], A. Sammak[2], M. Rimbach-Russ[1], G. Scappucci[1] & L. M. K. Vandersypen[1✉]

The scalability and power of quantum computing architectures depend critically on high-fidelity operations and robust and flexible qubit connectivity[1–3]. In this respect, mobile qubits are particularly attractive as they enable dynamic and reconfigurable qubit arrays. This approach allows quantum processors to adapt their connectivity patterns during operation, implement different quantum error correction codes on the same hardware and optimize resource use through dedicated functional zones for specific operations such as measurement or entanglement generation[4–7]. Such flexibility also relieves architectural constraints, as recently demonstrated in atomic systems based on trapped ions[4,5] and neutral atoms manipulated with optical tweezers[6,7]. In solid-state platforms, highly coherent shuttling of electron spins was recently reported[8,9]. A key outstanding question is whether it may be possible to perform quantum gates directly on the mobile spins. Here we demonstrate two-qubit operations between two electron spins carried towards each other in separate travelling potential minima in a semiconductor device. We find that the interaction strength is highly tunable by their spatial separation. When we shuttle the two spins towards the centre by 120 nm each for a total displacement of 240 nm, we achieve an average two-qubit gate fidelity of about 99%. Furthermore, we implement conditional post-selected quantum state teleportation between qubits separated by 320 nm with an average gate fidelity of 87%, showcasing the potential of mobile spin qubits for non-local quantum information processing. We expect that operations on mobile qubits will become a universal feature of future large-scale semiconductor quantum processors.

Quantum computing offers the promise to solve complex problems that are intractable for classical computers. As quantum processors scale up, maintaining high connectivity between qubits becomes crucial for implementing effective error correction schemes[1–3]. However, traditional architectures are often restricted to interactions between nearest neighbours, constraining the options for quantum error correction codes and potentially increasing overhead. Mobile qubits offer a promising alternative by enabling flexible connectivity between qubits, thus reducing the overhead associated with error correction schemes[4–7].

Among the various quantum computing platforms, gate-defined semiconductor spin qubits[10] have emerged as a promising candidate. These qubits offer a compelling combination of extended coherence times[11], high-fidelity operations[12–20], compatibility with established semiconductor manufacturing techniques[21–25] and the potential for high-temperature operation[26,27].

Inspired by mobile qubit approaches in atomic systems, in which trapped ions[4,5] and neutral atoms[6,7] have demonstrated the power of reconfigurable qubit arrays, the question arises whether mobile semiconductor spin qubits offer a route to realize flexible connectivity in a solid-state platform. In recent experiments, a travelling-wave

potential, generated by phase-shifted sinusoidal signals applied to successive gate electrodes, transports the spin qubit within a moving quantum dot[8,28–30]. With this so-called conveyor shuttling method, a 99.5% fidelity was achieved when shuttling across an effective 10-μm distance in less than 200 ns (ref. 9).

Building on these advances, we foresee scalable mobile spin qubit architectures based on conveyor-mode shuttling as shown in Fig. 1a, in which qubits can be selectively transported between storage zones and interaction regions are formed by pairs of independent conveyor channels. This approach enables efficient resource sharing and flexible qubit connectivity through shared control lines and sparse storage zones. It may also facilitate uniformly high gate fidelities by performing operations in optimized local electromagnetic environments. The fundamental building block and key challenge for realizing such architectures is the ability to precisely control the interaction between two mobile spin qubits by selectively bringing them together using independent conveyor channels and to achieve high gate fidelities[31].

In this work, we focus on demonstrating this critical operation using a linear array implementation. We simultaneously shuttle two spin qubits in conveyor mode towards each other, such that their wavefunctions

[1]QuTech and Kavli Institute of Nanoscience, Delft University of Technology, Delft, The Netherlands. [2]QuTech and Netherlands Organisation for Applied Scientific Research (TNO), Delft, The Netherlands. [3]These authors contributed equally: Y. Matsumoto, M. De Smet. ✉e-mail: L.M.K.Vandersypen@tudelft.nl

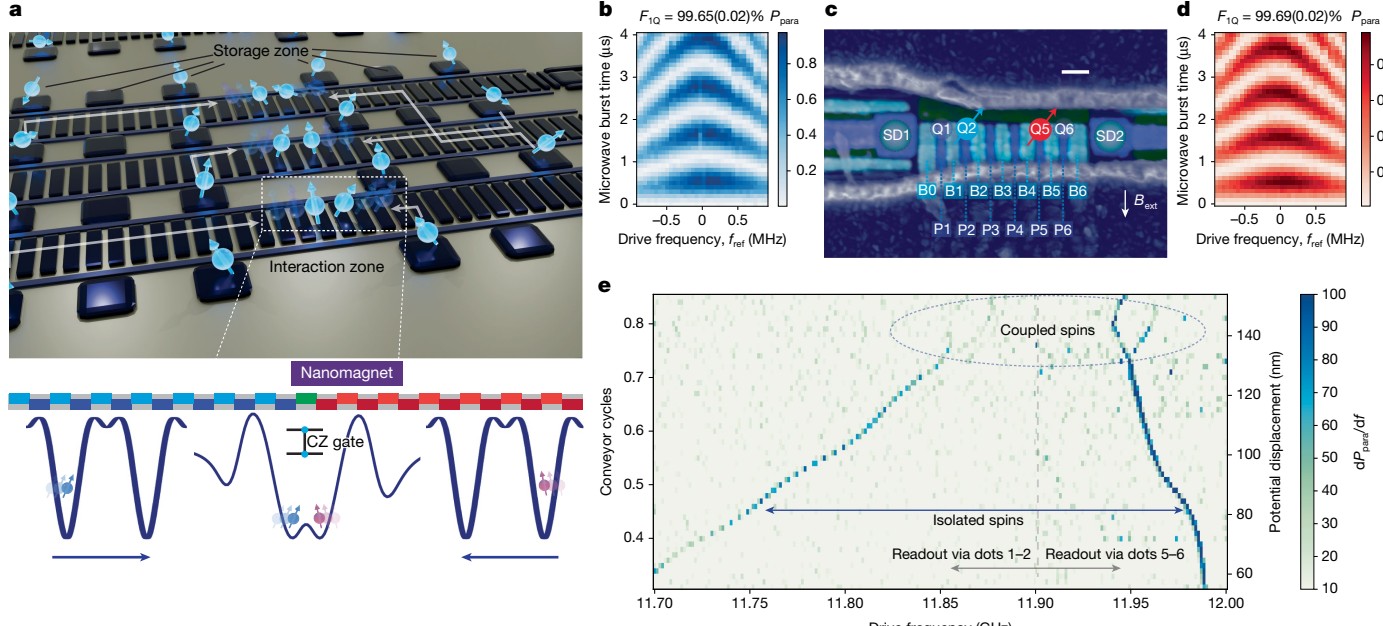

**Fig. 1 | Mobile spin qubits and shuttling-based architecture. a**, Conceptual architecture for a scalable mobile spin qubit processor based on conveyor-mode shuttling. Qubits can be transported between static storage zones (static dots) and pairs of adjacent conveyor channels that meet at shared interaction zones. Two-qubit operations are performed by simultaneously shuttling two qubits inside the same channel to an interaction zone. This design enables efficient resource sharing. Vertical transport between parallel conveyor channels, passing through vacant storage zones, allows for flexible connectivity across the array (see white arrows). Readout zones with sensing dots and ancilla qubits at array end points enable parallel measurements of several qubits. **b,d**, Parallel spin probability plotted as a function of drive frequency detuning from $f_{ref}$ (11.657 GHz for Q2 and 11.999 GHz for Q5) and microwave burst time reveals Rabi chevron patterns from single-qubit rotations for Q2 and Q5. The average single-qubit gate fidelities, extracted by randomized benchmarking, are shown at the top of the respective chevron patterns. **c**, False-coloured scanning electron microscopy image of a nominally identical device to the one used in this work. The colours indicate different metallization layers. Six plungers (P, blue), seven barriers (B, cyan) and two screening gates (dark green) form two distant double dots 1–2 and 5–6 (indicated by numbered circles). Two sensing dots (SD) are placed at both ends of the array. A cobalt micromagnet, shown in dark blue, is placed on top of the active area. Scale bar, 100 nm. **e**, EDSR spectroscopy as a function of microwave drive frequency and the number of conveyor cycles (left axis), with the corresponding nominal displacement of the potential minima (right axis). The dashed vertical line at 11.90 GHz indicates the boundary between two separate measurements. Between 11.70 and 11.90 GHz, the plot shows the numerical derivative of the parallel spin probability of Q1 and Q2, with Q5 initialized in a mixed state. Between 11.90 and 12.00 GHz, the derivative of the parallel spin probability of Q5 and Q6 is shown, with Q2 initialized in a mixed state.

begin to overlap and a two-qubit exchange interaction is activated. By systematically examining the exchange interaction over various shuttle distances and barrier voltage configurations, we seek to identify the conditions under which tunable and high-fidelity two-qubit gates can be realized. Moreover, we explore a regime in which merging extremely elongated quantum dot potentials may lead to exchange interaction saturation[32-34]. Furthermore, we use the two-qubit gate on mobile spins to entangle and separate two spins and use this entanglement to realize conditional post-selected quantum state teleportation of single-spin states, effectively spanning five quantum dots.

## Mobile spin qubits and shuttling-based architecture

We operate mobile spin qubits in a six-quantum-dot array fabricated on an isotopically purified $^{28}$Si/SiGe heterostructure featuring a 7-nm quantum well[35] (Fig. 1c). The quantum dots are defined by three layers of Ti:Pd gates separated by $Al_2O_3$, consisting of screening, plunger (P) and barrier (B) gates. Details on the device fabrication can be found in Supplementary Information section A. For readout, sensing dots are positioned at both ends of the array. A cobalt micromagnet integrated on top of the device provides the magnetic field gradient necessary for single-qubit control by means of electric-dipole spin resonance (EDSR)[36]. We operate the device in an in-plane magnetic field of 260 mT at a temperature of 200 mK (ref. 26).

To test two-qubit operations on mobile qubits, we will load spins Q2 and Q5 from dots 2 and 5 into two independent conveyor potentials

travelling inside the channel in between. Spins in dots 1 and 6, Q1 and Q6, serve as stationary readout ancillas. Readout relies on a variant of Pauli spin blockade, which reveals the parity of Q1–Q2 and Q5–Q6. The coherence times and control fidelities are summarized in Supplementary Figs. 5 and 6.

For conveyor-mode shuttling, several phase-shifted sinusoidal signals are applied to plungers and barrier gates. As in ref. 9, two tones are applied, with frequency $f$ ($f/2$) and a spatial period of four (eight) gate electrodes. We adjust the DC gate voltages in the channel such that the background potential landscape is approximately flat. As a result, a travelling-wave potential carries electrons within a moving potential well. This method allows for continuous control of the electron positions along the array. In this work, we operate two mobile qubits using two moving potentials, which move from the positions of dot 2 and dot 5 towards the centre. This is achieved by applying sine waves with phase offsets that increase symmetrically from P2 and P5 towards the central barrier B3 (Supplementary Table II).

To characterize the simultaneous transport of two mobile spin qubits, we perform EDSR spectroscopy as a function of the number of conveyor cycles applied (Fig. 1e), in which one conveyor cycle corresponds to a nominal displacement of 180 nm, which is twice the plunger gate pitch (we refer to the number of conveyor cycles of the primary frequency component $f$). In the left part of the plot, Q2 is initialized in the spin-down state ($\equiv |0\rangle$), whereas Q5 is prepared in a random state; in the right part of the plot, Q2 is prepared in a random state, whereas Q5 is initialized in the spin-up state ($\equiv |1\rangle$). For each

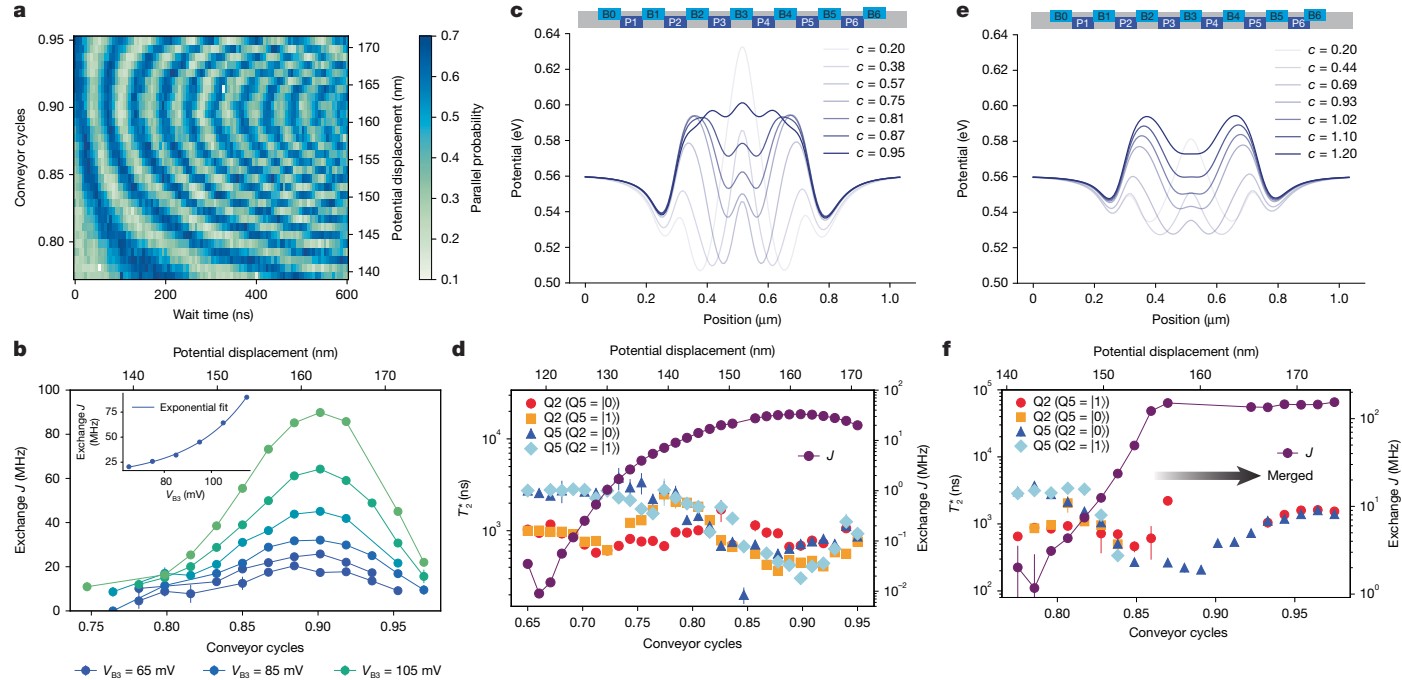

**Fig. 2 | Exchange and coherence of mobile spin qubits. a**, Two-dimensional map showing DCPhase oscillations as a function of wait time and conveyor cycles (left axis), with corresponding nominal potential displacement (right axis), with a 65-mV pulsed voltage offset on B3. See main text for further details. **b**, Exchange coupling $J$ versus conveyor cycles (bottom axis), with corresponding nominal potential displacement (top axis), at different pulsed voltage offsets on $V_{B3}$ and extracted from DCPhase oscillations as in panel **a**. The inset shows the peak exchange strength versus the voltage on B3 and an exponential fit. **c**, Simulated potential profiles showing the evolution of the two moving potential minima during shuttling, in which $c$ represents the number of conveyor cycles applied. A cycle is defined by one full period of the primary frequency component $f$. **d**, Measured dephasing times $T_2^*$ for both qubits under different states of the other spin: Q2 (Q5 = |0⟩), Q2 (Q5 = |1⟩), Q5 (Q2 = |0⟩) and Q5 (Q2 = |1⟩), plotted against conveyor cycles (bottom axis), with corresponding nominal potential displacement (top axis). The right axis shows the exchange coupling $J$ extracted from the measured difference in Ramsey frequency depending on the state of the other spin. A 9.5-mV pulsed offset is applied to B3, consistent with panel **c**. **e**, Simulated potential profiles at different conveyor cycles $c$ for the elongated dot configuration. As the confinement potential evolves with increasing $c$, the elongated potential minima merge into a single even more elongated potential, supporting strong Coulomb interactions between the electrons, which favours the formation of a Wigner molecule. **f**, Exchange coupling and dephasing times $T_2^*$ for Q2 (Q5 = |0⟩), Q2 (Q5 = |1⟩), Q5 (Q2 = |0⟩) and Q5 (Q2 = |1⟩) in the merged configuration of panel **e**, showing exchange saturation at larger conveyor cycles. The exchange coupling is extracted from DCPhase oscillations in this condition.

configuration, we shuttle both spins using conveyor-mode shuttling, apply a microwave burst for EDSR, return the spins to dot 2 and dot 5 and perform readout. The observed shift in resonance frequency with increasing conveyor cycles indicates a gradual spin displacement along the channel. The different slopes for Q2 and Q5 arise from the different magnetic field gradients experienced by the qubits as they are shuttled towards the centre. As the electrons approach each other, the EDSR spectra reveal split lines, indicating that the two spins interact, such that the resonance frequency of one is dependent on the state of the other.

## Exchange and coherence of mobile spin qubits

Next we evaluate the controllability of the exchange interaction achieved through shuttling. To characterize the exchange interaction, we use a decoupled controlled-phase (DCPhase) sequence: after initialization, an $R_x(\pi/2)$ pulse is applied to Q2, followed by shuttling both qubits together to activate exchange coupling for a duration $t/2$ and shuttling the spins back to their starting position. $R_x(\pi)$ gates are then applied to both spins in their respective dots, followed by another exchange interaction period $t/2$. Finally, another $R_x(\pi/2)$ pulse is applied to Q2 before measuring both qubits using parity readout against their reference spins. Figure 2a presents the measured DCPhase oscillations versus conveyor cycles with a 65-mV pulsed offset applied to barrier gate B3. The oscillation frequency, which directly reflects the exchange amplitude $J$, increases smoothly as the potential minima

are moved towards each other before plateauing and subsequently decreasing.

Figure 2b shows the estimated exchange strength from DCPhase measurements performed at different B3 pulsed offsets (ranging from 65 to 115 mV). For each curve, $J$ reaches a maximum and then slightly decreases. This is counterintuitive in the picture, in which two potential minima progressively move towards each other. It results from the fact that the central barrier gate B3 receives a lower conveyor amplitude (and a negative pulsed offset) compared with the other conveyor gates (Supplementary Information sections C and D). This keeps the minima from the two conveyor channels separated by a tunnel barrier while still allowing a controlled exchange coupling. Figure 2c illustrates this through simulated potential profiles at different conveyor cycles $c$. As seen in the inset to Fig. 2b, $J$ increases exponentially with the pulsed offset on B3, consistent with the Fermi–Hubbard model describing tunnel-coupled quantum dots[37]. At the start of the conveyor (zero conveyor cycles), $J$ is too small to measure. The exchange strength between the mobile qubits can be tuned up to 90 MHz through control of both the number of conveyor cycles $c$ and the barrier voltage. Although higher exchange strengths are achievable, they were difficult to measure owing to their rapid decay under these conditions.

The two-qubit gate fidelity that can be achieved depends on the balance between $J$ and $T_2^*$. Figure 2d presents the $T_2^*$ dephasing times of Q2 and Q5, along with the corresponding exchange strength $J$, as a function of the number of conveyor cycles. As $c$ increases, leading to a stronger

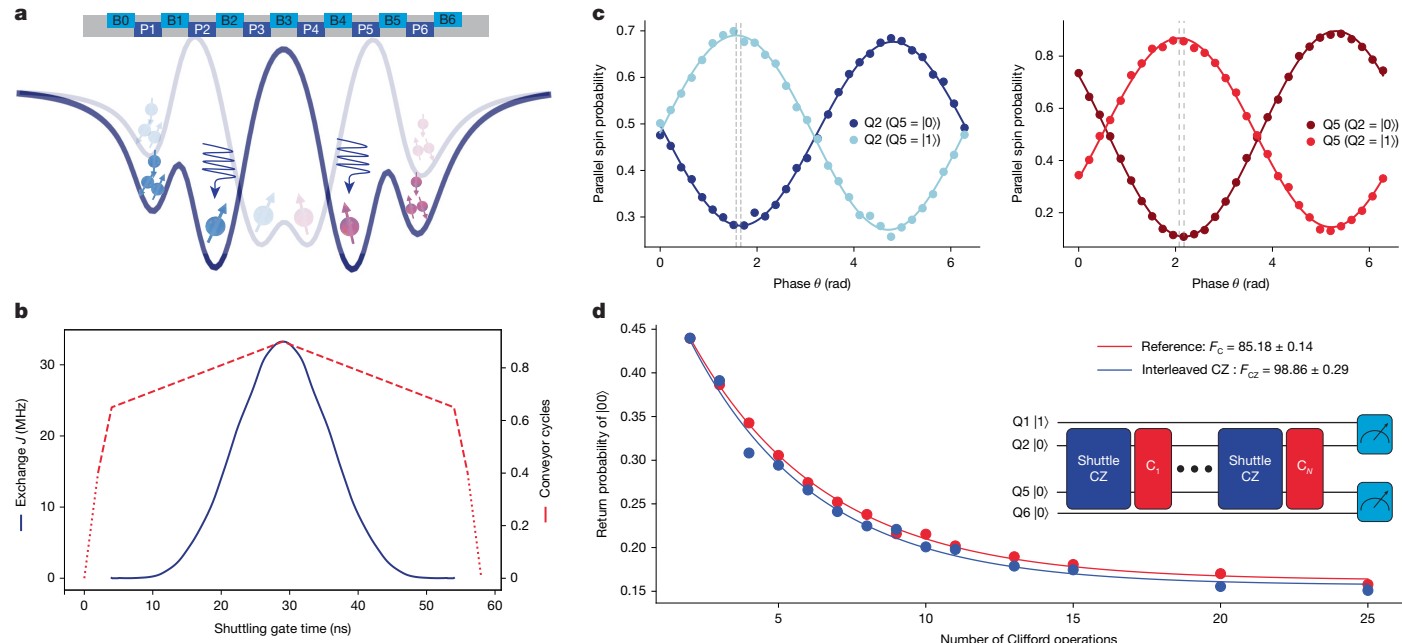

**Fig. 3 | Fidelity benchmarking of shuttling-based CZ gate. a**, Schematic illustration of the gate electrodes and simulated potential profiles with static dots (dark line) and the conveyor during the interaction phase (faint line). **b**, Time evolution of the exchange coupling strength $J$ (left axis) and the nominal potential displacement expressed in conveyor cycles (right axis) throughout the CZ gate operation. The dotted lines indicate the loading and unloading phases between the initial positions of the conveyor's potential minima and the static dots. The pulse shape is designed to achieve adiabatic control of the exchange interaction. **c**, Calibration measurements of the CZ gate showing parallel spin probability oscillations for Q2 (left) and Q5 (right) as a function of an applied virtual phase shift $\theta$. The measurements are performed with the other qubit initialized in either the $|0\rangle$ or the $|1\rangle$ state, as indicated. **d**, Results of interleaved randomized benchmarking, showing the return probability versus the number of Clifford operations for both reference (red) and interleaved CZ gate (blue) sequences. The schematic illustrates the interleaved randomized benchmarking protocol, in which shuttling-based CZ gates ('Shuttle CZ') are interleaved between random Clifford operations ($C_1$ to $C_N$) for the interleaved measurements, whereas reference measurements are performed with only the red Clifford operations.

exchange interaction and faster gate operation, we observe a decrease in $T_2^*$ for both qubits. The difference in $T_2^*$ depending on the state ($|0\rangle$ or $|1\rangle$) of the other qubit can be attributed to the spatial magnetic field gradient in the device[15]. We note that $T_2^*$ during shuttling in the device typically exceeds the static $T_2^*$ measured at fixed positions along the channel[9]. This is because of motional averaging, as the qubit samples different local environments at a rate faster than the correlation time of both the nuclear field fluctuations and charge noise[8,38]. On the basis of this characterization, we identify an operating regime that balances a strong enough exchange coupling for fast gates while maintaining sufficient coherence times. This condition, corresponding to 0.9 conveyor cycles and an exchange coupling of 33 MHz, enables the high-fidelity two-qubit operations demonstrated in Fig. 3.

Before proceeding to quantifying the gate fidelity, we investigate the exchange strength and spin coherence in a configuration in which two elongated travelling potential minima are created by applying a single sinusoidal wave having a spatial period corresponding to eight gate electrodes (once again with phase offsets that increase symmetrically from the outer gates towards the centre; here B3 receives a pulse offset of −10 mV). When the two elongated potentials overlap substantially at the centre, the system transitions away from a double-dot regime in which the Fermi–Hubbard description is applicable. Numerical simulations (Fig. 2e) reveal how the potential profile transitions from a barrier-controlled double-well configuration into a single highly elongated potential.

Figure 2f shows that, in this regime, $J$ initially exhibits the standard exponential increase expected for a double quantum dot (up to approximately 0.86 conveyor cycles), in which the potential barrier dictates the coupling. Beyond this point, as the two elongated potentials merge, $J$ no longer increases exponentially but instead saturates. This saturation may originate from strong electron–electron interactions within the merged elongated dot[32–34]. Notably, once in this merged regime, $J$ becomes relatively insensitive to barrier voltage variations, enabling an enhancement of $T_2^*$ while still maintaining substantial exchange coupling. Unfortunately, before the conveyor has reached this favourable condition, it passes through a region in which $T_2^*$ is very short, resulting in overall low two-qubit gate fidelities. In fact, when measuring Q2's $T_2^*$ with Q5 initialized in $|0\rangle$, or Q5's $T_2^*$ with Q2 in $|1\rangle$, we cannot even accurately estimate $T_2^*$ because coherence is lost rapidly before $J$ reaches the saturation regime. In the region with very short $T_2^*$, the growing electric dipole increases sensitivity to charge noise and, also, the frequency shifts because both the magnetic field gradient and the exchange interaction add constructively, as voltage fluctuations or charge noise simultaneously affect both the Zeeman energy splitting and the exchange interaction in a correlated way[15]. Notably, in this configuration of interacting mobile spins, tuning the exchange strength does not require independent dynamic control of the central barrier (B3) potential. Instead, the barrier gate is part of the travelling-wave potential. Although a static barrier voltage can be used in practice to adjust the exchange coupling, comparable exchange values can, in principle, be obtained only by adjusting the shuttling distance.

## Fidelity benchmarking of shuttling-based CZ gate

Returning to the conveyor configuration of Fig. 1a,c, we evaluate the performance of a shuttling-based conditional-Z (CZ) gate. We set the target maximum exchange strength of 33 MHz such that the exchange strength is much smaller than the Zeeman energy difference between Q2 and Q5 after they have been brought into each other's vicinity. The Zeeman energy difference suppresses the flip-flop terms of the exchange interaction, leaving only the ZZ term active.

**a**

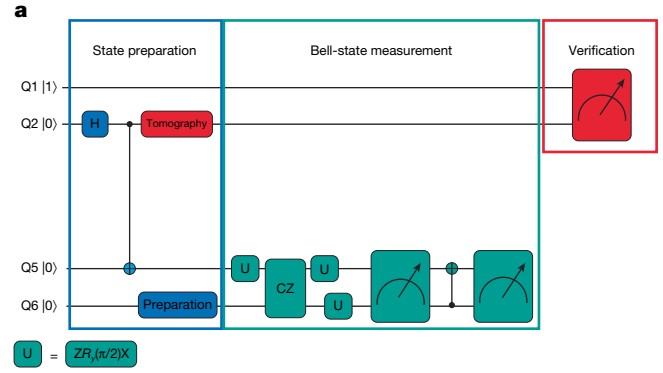

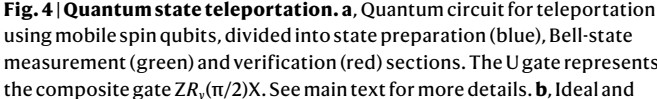

**b**

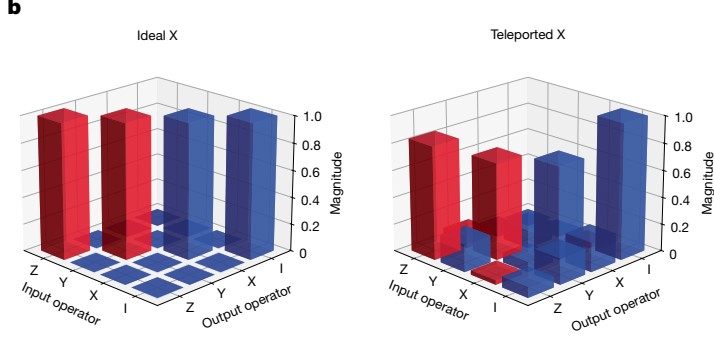

**Fig. 4 | Quantum state teleportation. a**, Quantum circuit for teleportation using mobile spin qubits, divided into state preparation (blue), Bell-state measurement (green) and verification (red) sections. The U gate represents the composite gate $ZR_y(\pi/2)X$. See main text for more details. **b**, Ideal and experimentally measured Pauli transfer matrices for the X operation, obtained through quantum process tomography of the teleportation channel. Red (blue) bars represent positive (negative) values of the magnitude shown.

Figure 3a illustrates schematically how the device is operated. The four qubits (Q1, Q2, Q5, Q6) are initialized in the |1000⟩ state, in which Q1 and Q6 serve once again as ancilla qubits for parity readout. Microwave bursts for single-qubit control are applied while qubits Q2 and Q5 are confined in the static dots 2 and 5. For the two-qubit CZ gate, conveyor-mode shuttling carries Q2 and Q5 towards each other and back for a controlled duration.

Figure 3b shows the target nominal potential displacement over time and the resulting $J$ throughout the two-qubit gate (interpolated from the data points in Fig. 2d). The pulse sequence is designed to balance speed and adiabaticity, consisting of the following main stages: first, a 2-ns loading phase (dotted lines) carries the electrons from static dot 2 (Q2) and 5 (Q5) to the initial conveyor potential at $c = 0.4$ (in which $c = 0$ would correspond to a conveyor minimum centred below P2 and P5); second, a fast 2-ns approach phase using a 125-MHz conveyor frequency, corresponding to a speed of 22.5 m s$^{-1}$, minimizes dephasing during initial transport (from $c = 0.4$ to $c = 0.65$); third, a 25-ns interaction phase using a 10-MHz conveyor frequency, corresponding to a speed of 1.8 m s$^{-1}$, for which the inter-qubit distance is adiabatically reduced to activate the exchange interaction; and finally, a symmetric return sequence. Including the 2-ns loading and approach phases, the total CZ gate time is 58 ns. The two spins are initially 270 nm apart, spanning four quantum dots.

The CZ gate calibration uses a set of measurements similar to those shown in Fig. 3c, exploring different centre barrier voltages (Supplementary Fig. 8). In the measurement sequence, one spin (Q2 for the left panel, Q5 for the right panel) is prepared in a superposition state using an $R_x(\pi/2)$ gate and the other spin is initialized in either |0⟩ or |1⟩. The spins are then shuttled to the centre, at which the exchange interaction is activated, followed by return shuttling to their initial positions. Before measurement, a virtual $R_Z(\theta)$ rotation is applied through microwave phase adjustment, followed by another $R_x(\pi/2)$ gate. The left panel shows the parallel spin probability for Q2 as a function of $\theta$, with the dark blue curve representing Q5 in |0⟩ and the light blue curve for Q5 in |1⟩. The right panel shows the complementary data for Q5, with the dark red curve representing Q2 in |0⟩ and the red curve for Q2 in |1⟩. The two cases are (nearly) out of phase, as expected for a properly calibrated CZ gate. These same measurements also serve to calibrate the single-qubit phase shifts accumulated during shuttling. This is done by varying $\theta$ such that the largest contrast between the light and dark data points is reached. Looking closely, in Fig. 3c, a slight deviation from a π controlled phase is seen even though the gate is properly calibrated. This arises from further phase accumulation owing to the extra microwave burst[26] that is applied only when preparing the other qubit in |1⟩, rather than from CZ gate imperfections (Supplementary Fig. 9).

Figure 3d shows the results of interleaved randomized benchmarking measurements. For every reference and interleaved measurement, we produce 120 distinct sequences at random. We performed 800 single shots for each sequence, with an average of approximately 250 single shots being post-selected[39]. The data are plotted as the return probability to the initial state |00⟩ versus the number of Clifford gates $N$, for both the reference sequence and the interleaved benchmarking sequence. The decay of the return probability is analysed to extract the fidelity of the Clifford gates $F_C$ and the fidelity of the CZ gate $F_{CZ}$. The estimated fidelity $F_{CZ} = 98.86 \pm 0.29\%$ demonstrates the high performance of the shuttling-based CZ gate. We discuss the error budget and the consistency of the various fidelities in Supplementary Information section E.

## Quantum state teleportation

Finally, to demonstrate that we can take advantage of spatially separated entangled spins created by the mobile spin-qubit approach, we implement a conditional post-selected quantum state teleportation protocol. Figure 4a illustrates the quantum circuit for teleportation using mobile spin qubits. The circuit is divided into three main sections: preparation of Q6 and Bell-state preparation of Q2 and Q5 through the controlled interaction of mobile spins (blue); a Bell-state measurement of Q5 and Q6 (green); and verification (red). The Bell-state measurement teleports the state of Q6 to Q2. However, the implemented Bell-state measurement cannot fully resolve all four Bell states owing to limitations in the parity readout, making the protocol conditional. Analogous to photonic experiments[40,41], we use post-selection on the measurement outcomes, which preserves the quantum nature of the protocol while succeeding probabilistically (Methods). We focus on the post-selected cases corresponding to the $\Psi^+$ Bell-state measurement outcome, which implements an effective X gate operation on the teleported state. We characterize the teleported state using quantum state tomography[42] of Q2 (experiments that separately test how well the polarization and phase of Q6 are teleported are presented in Supplementary Information section F).

Figure 4b presents the ideal (left) and experimental (right) Pauli transfer matrices for this X operation, corresponding to the even + even (|00⟩) parity readout outcomes between Q5 and Q6. The experimental Pauli transfer matrices are reconstructed using least-squares optimization with complete positivity and trace-preserving constraints to ensure the physical validity of the estimated quantum process[43,44]. After correcting for state preparation and measurement errors, we determine that the average fidelity of the X gate is $86.7 \pm 0.9\%$, which corresponds to the fidelity of the teleportation process (for the $\Phi^-$ measurement outcome, the teleportation fidelity is lower). The uncertainty is the

standard deviation obtained through bootstrap resampling. Supplementary Information section G quantifies the various error contributions. The extracted fidelity greatly exceeds the classical bound of 2/3, demonstrating that this protocol achieves genuine quantum state teleportation.

## Conclusion

In this work, we demonstrate the concept of two-qubit gates between mobile semiconductor spin qubits. The exchange interaction is activated and precisely controlled in time by moving two electrons towards each other in the minima of two travelling conveyor potentials, which allows for a two-qubit CZ gate fidelity of about 99%. The observed saturation of the exchange coupling in elongated conveyor potential minima, possibly because of the formation of strongly correlated electron states, could provide an interesting avenue for implementing robust two-qubit gates.

Moreover, we showcase the potential of distributed quantum computing through conditional post-selected quantum state teleportation between distant qubits. In future quantum processors, crucial operations such as magic-state distillation could be performed in dedicated regions and the resulting states could be distributed to computational zones through teleportation, enabling efficient resource sharing across the processor. The next step will be to demonstrate deterministic teleportation by incorporating fast, non-demolition readout techniques that enable real-time feedforward operations based on four distinguishable Bell-state measurement outcomes[45–47].

Notably, the two-qubit gate realized here differs fundamentally from conventional exchange operations between static spins in predefined dots and provides several key advantages over making spins hop towards each other through a chain of static dots (see also Supplementary Information section B): (1) the use of shared control lines in conveyors greatly reduces the wiring overhead and complexity; (2) it allows for fast high-fidelity two-qubit gates between spins that are initially far apart; and (3) control of the conveyor velocity can be used to implement in one step a CZ-like, CX-like or SWAP-like operation[48], offering a path towards dynamically programmable two-qubit logic.

Scaling up quantum processors based on mobile spin qubits would require advances along three key directions. First, given the individual DC tuning of each gate in this work, an important next step is to demonstrate qubit transport through long-distance conveyor channels with shared control lines[28,30,49]. Second, as illustrated in Fig. 1a, implementing arrays of storage zones connected by shared conveyor belts to exploit flexible connectivity between many qubits. Third, realizing simultaneous shuttling operations in parallel conveyor channels to increase the number of mobile qubits that can be controlled independently.

These advances, building on the demonstration of high-fidelity two-qubit operations and quantum state teleportation with mobile spin qubits shown here, represent vital steps towards large-scale, reconfigurable quantum processors.

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

## Methods

### Parity readout and initialization sequence

Here we describe the details of the initialization sequence. To optimize the readout visibility through Pauli spin blockade, we use the (3,1)–(4,0) charge transition for dots 1–2 and the (1,3)–(0,4) transition for dots 5–6. First, we check the parity of the (Q1,Q2) and (Q5,Q6) pairs. If the parity is even, we apply an X gate to Q2 or Q5 using microwave-driven feedback control in the (3,1) and (1,3) charge states, respectively. We then remeasure the parity of the (Q1,Q2) and (Q5,Q6) pairs and post-select for odd parity. Subsequently, we adiabatically transition the (Q1,Q2) and (Q5,Q6) pairs from (4,0) to (3,1) and from (0,4) to (1,3) charge states, respectively, over a duration of 50 ns. To enhance adiabaticity during this process, we apply pulses of approximately +200 mV to the inter-dot barrier gates (B1 and B5) compared with the readout conditions. When this adiabatic initialization is successful, the spin states of (Q1,Q2) and (Q5,Q6) become (1,0) and (1,0), respectively, as determined by the Zeeman energy differences between each pair. Finally, we apply an X gate to Q5, initializing the spin states of (Q1,Q2,Q5,Q6) to (1,0,0,0).

### Two-tone conveyor pulse

Following previous work[9], we use a combination of two sine waves for each gate: one at frequency $f$ and another at $f/2$, which can be expressed as:

$$V_n(t) = V_n^{DC} + \frac{A}{2}[\sin(2\pi ft - \phi_n) + \sin(\pi ft - \theta_n)],$$

in which $V_n^{DC}$ represents the individual pulsed offset, $A$ is the amplitude and $\phi_n$ and $\theta_n$ are the phase offsets for the respective frequency components. This approach creates wider potential barriers between neighbouring conveyor minima, which greatly reduces the probability of charge leakage during transport.

### Interleaved randomized benchmarking

We analyse the average gate fidelity of the shuttling-based CZ gate using interleaved randomized benchmarking. In our analysis, the sequence fidelity $F_t(L)$ was measured as $P_{\downarrow\downarrow}(L)$, which is the probability that both qubits are in the spin-down state after applying $L$ Clifford gates. The measured sequence fidelity $F_t(L)$ is modelled as:

$$F_t(L) = A_t p_t^L + B_t \qquad (1)$$

in which $A_t$ and $B_t$ are constants accounting for state preparation, measurement errors and any residual offsets and $p_t$ is the depolarizing parameter. The fidelity of the Clifford gates $F_C$ is then calculated from $p_t$ using:

$$F_C = \frac{1 + 3p_t}{4} \qquad (2)$$

For the CZ gate fidelity $F_{CZ}$, the sequence fidelity is measured with CZ gates interleaved between Clifford gates. The CZ gate fidelity is then estimated by comparing the decay rates of the sequence fidelities with and without the interleaved CZ gates:

$$F_{CZ} = \frac{1 + 3p_{CZ}/p_{ref}}{4} \qquad (3)$$

in which $p_{ref}$ is the depolarizing parameter obtained from the reference (non-interleaved) benchmarking sequence.

### Detailed teleportation protocol

For Bell-state preparation, we first perform single-qubit operations with Q2 and Q5 in dots 2 and 5, then shuttle them together to implement a CZ operation and finally separate them to their original positions to create a distributed entangled state between distant qubits. The Bell-state measurement is achieved through a series of operations: first, a measurement basis transformation is applied to map the four Bell states onto the four computational basis states ($|00\rangle$, $|01\rangle$, $|10\rangle$, $|11\rangle$) of Q5 and Q6. Then, two sequential parity readouts, separated by a CNOT gate, are performed. The CNOT between Q5 and Q6 is implemented by first applying a $R_y(\pi/2)$ gate to Q6, followed by a CZ operation achieved by pulsing a barrier gate between static dots to control the exchange interaction and finally another $R_y(-\pi/2)$ gate to Q6. It is important to note that, owing to the mixing of the triplet $T_0$ and the singlet $S$ during Pauli spin blockade parity readout[39,50], this method can distinguish between the $|00\rangle$ and $|11\rangle$ states but it cannot differentiate between the $|10\rangle$ and $|01\rangle$ states. This limitation affects the ability to fully discriminate between all four Bell states, hence we here achieve conditional quantum state teleportation. Alternative measurement schemes will be necessary to overcome this constraint.

For verification, we apply a tomography pulse on Q2 followed by parity readout of Q2 against the reference qubit Q1. The application of tomography pulses to Q2 before the Bell measurement is justified by the structure of the quantum teleportation protocol. Once the initial Bell pair is created between Q2 and Q5, the subsequent tomography pulses on Q2 and Bell measurement between Q5 and Q6 are independent events—their order does not affect the teleported state. This approach is particularly advantageous in the implementation in this work in which measurement times (40 µs) exceed coherence times, as it minimizes the delay between state preparation and the state projection by tomography pulses. Similar to photonic teleportation experiments[40,41], we use post-selection, which, although reducing the overall success probability, preserves the quantum nature of the protocol.

## Data availability

The raw measurement data and the analysis supporting the findings of this work are available in a Zenodo repository (https://doi.org/10.5281/zenodo.15052065)[51].

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

**Acknowledgements** We thank S. G. J. Philips for writing the control libraries and designing the printed circuit board, R. Schouten, R. Vermeulen, O. Benningshof and T. Orton for support with the measurement set-up and dilution refrigerator, B. Undseth for discussions on scalable architectures, X. Xue for discussions on the tuning of two-qubit gates, C. Wang for discussions on benchmarking and other members of the Vandersypen, Veldhorst, Scappucci and Dobrovitski groups for fruitful discussions. We acknowledge financial support from the Army Research Office (ARO) under grant number W911NF2310110. M.R.-R. acknowledges support from the Netherlands Organization of Scientific Research (NWO) under Veni grant no. VI.Veni.212.223 and by the EU through H2024 QLSI2. The views and conclusions contained in this document are those of the authors and should not be interpreted as representing the official policies, either expressed or implied, of the ARO or the US Government. The US Government is authorized to reproduce and distribute reprints for government purposes notwithstanding any copyright notation herein.

**Author contributions** Y.M. and M.D.S. performed the experiments and data analysis. Simulations were carried out by Y.M. Libraries for experimental control were written by S.L.d.S. and Y.M. Y.M., M.D.S., M.R.-R. and L.M.K.V. contributed to data interpretation. L.T. fabricated the device and S.V.A. refined the device design. A.S. and G.S. designed and grew the heterostructure. Y.M., M.D.S. and L.M.K.V. wrote the manuscript, with comments by all authors. Y.M. conceived the project. L.M.K.V. supervised the project.

**Competing interests** The authors declare no competing interests.

**Additional information**
**Correspondence and requests for materials** should be addressed to L. M. K. Vandersypen.
