## [Peer Review file · Nature]

Two-qubit logic and teleportation with mobile spin qubits in silicon

Corresponding Author: Professor Lieven Vandersypen

Version 0:

Reviewer comments:

Referee #1

(Remarks to the Author)

Review of the manuscript “Two-qubit logic and teleportation with mobile spin qubits in silicon” by Y. Matsumoto et al., submitted to Nature.

In this work, the authors first demonstrate and benchmark a controlled-phase (CPHASE) gate between two spin qubits in Si/SiGe, and in the second part of the manuscript, they present conditional post-selected quantum state teleportation.

The CPHASE gate between two spin qubits, each encoded in a single electron spin, has been demonstrated previously (e.g., Ref. 13). Conditional teleportation of quantum-dot spin states in Si/SiGe has also been reported [<https://doi.org/10.1038/s41467-020-16745-0>]. What is new here is that the two electrons—each representing a spin qubit—are mobile and are first shuttled toward each other using conveyor-belt-style shuttling, as described in Refs. [28–30]. Conveyor-mode shuttling is an attractive approach for qubit connectivity, as it appears to require a low number of control signals and can reduce cross-talk.

The combination of such shuttling with a reliable two-qubit gate has not been demonstrated before, and this represents a significant step toward more scalable spin qubit architectures, reconfigurable qubit arrays, and more flexible quantum error correction schemes. It is worth noting that combining already demonstrated spin-qubit functionalities—such as the CPHASE gate and shuttling—while maintaining high fidelity is a non-trivial task, and the authors have achieved this very effectively.

In the second part of the manuscript, the authors go one step further by integrating the shuttling and CPHASE gate into a conditional quantum teleportation protocol over a distance of four quantum dots. The protocol is primarily limited by unoptimized detection fidelity.

Beside the impressive experimental demonstration, the manuscript stands out by benchmarking fidelities and data analysis and transparent presentation of the data. Many details are given in the methods and supplementary information. It is also highly appreciated that the authors explore the regime of strong electron-electron interaction and saturation of exchange interaction, although this analysis reveals some unfortunate region.

This paper is a highlight on spin-qubit quantum computing. A publication in Nature is definitely recommended.

I would like to invite the authors to address the following items:

Most important:

1. Fine adjustments of gates: How are the DC offsets and phase offsets adjusted (Tables I to III)? How important is this adjustment? If the adjustment is significant, this should be discussed in the conclusion. It seems that these adjustments are still a complication which is worth to be solved in the future.
2. The “unfortunate” low T_2^* region is very interesting. Do I understand correct that the regime in Fig. 2e should be better in theory, but cannot be exploited due to this unfortunate region? Is the elongated QD more exposed to dephasing? What is

- known about this unfortunate region? Do valley, orbital excitations play a role? How can such a region be avoided?
3. By comparing the two CHPASE regimes in Fig2, is there an optimal or just better gate pitch to further exploit the strong electron-electron regime? Or can the CPHASE gate be performed with any reasonable gate-pitch?
 4. Please specify the shuttle distances and the quantum teleportation distance in the abstract.

Minor:

5. Does the shuttling (125 MHz with high amplitude) contribute to the heating? Would it contribute if shuttling is across longer distances?
6. If you have information about the valley splitting of the device, please add it to the methods/supplements.

Referee #2

(Remarks to the Author)

The work by Y. Matsumoto reports on two-qubit logic for silicon spin qubits in a unique scheme. The experiments are elaborated and the results are enlightening. The authors furthermore claim its substantial implication for future silicon-based quantum processors, which is of high relevance to a broad readership. The manuscript is certainly publishable in a high-profile journal, but my view is that its impact - whether it presents the level of an advance expected for a Nature paper - hinges critically on the benefits of the proposed reconfigurable architecture for large-scale integration based on the demonstrated two-qubit gate operation.

The main point:

1) I find it hard to see from the manuscript how the authors propose to implement the mobile-spin two-qubit logic with finite sets of independent conveyor potentials in a large-scale system. What is the estimated required number of phase-shifted sinusoidal signals (not the tones) in the entire qubit array, in principle, for a reconfigurable, flexibly connected 2-dimensional qubit array of $n \times m$ - where n is the number of conveyor rows and m is the number of interaction zones along a conveyor row? Can the authors illustrate the idea e.g. in Fig. 1a?

Other points:

- 2) How are the potentials shown in Figures 2c and e simulated? The simulated potential seems to be always symmetric - does the so-called symmetric operation comes for free in this two-qubit gate scheme?
- 3) In the present study, the two-qubit fidelity is about 99%. What would be key in further improving the fidelity in this scheme?
- 4) Can the authors explain the expected key difference of the presented architecture (Fig. 1a) in terms of the electronics and wiring resources for high-fidelity two-qubit gates from the other existing schemes, where the two-qubit logic that demands for most precise tuning are handled by dedicated gate electrodes?
- 5) What will be the impact of the pulse sequence of a CZ gate (shown in Fig. 3b) to other qubits that are being transported in conveyor channels?
- 6) The authors conclude in the supplementary section that somehow the teleportation process fidelity depends rather strongly on the Bell measurement outcome. In my opinion, this should be mentioned in the main text.

Minor comments and suggestions.

- 7) For improved readability, please specify somewhere that the qubit 0 state corresponds to the spin-down state in this work.
- 8) In the caption of Fig. 1(c), the color assignment seems swapped.
- 9) I suspect that the authors meant a " $Ry(-\pi/2)$ gate" by a " $Ry(\pi/2)$ gate" (used in Sec. II D).
- 10) I suspect that the section title of Methods sec. B contains a typo ("tone" instead of "tune").
- 11) In supplementary Tables I - III, some phase offsets exceed 2π - can the authors double check?
- 12) Supplementary Table III is not referred to in the main text when appropriate. In addition, it would be helpful to specify for which dataset the configuration is used in the table caption.
- 13) Can the authors double check the description of the error bars in the caption of Fig. 7?
- 14) In Fig. 8, is the AC voltage offset the same as the DC offset?
- 15) In Fig. 8, what do the error bars mean in the last sentence of the caption? Should it be deleted?
- 16) In Supp. Sec. E, the qubit frequencies are wrongly labeled, if I am not mistaken.

Referee #3

(Remarks to the Author)

In the manuscript titled "Two-qubit logic and teleportation with mobile spin qubits in silicon", the authors implemented a protocol in which they performed transportation and coupling of two electron spins through a single pulse sequence. After demonstrating the effectiveness of this protocol, the authors performed a partial demonstration of quantum teleportation in such a system. I find the results of both sets of experiments quite impressive, with a controlled-Z gate fidelity of nearly 99% and the teleportation fidelity of 87%. While I have no doubt that the authors have made significant progress in their ability to control electron spins in Si, I am still ambivalent on whether the advances reported in this manuscript is worthy of a Nature article, as I discuss below.

My criterion for the importance/impact of an article is based on what are represented by past Nature and Science articles in the field of spin qubits in semiconductors. What I found from those articles are usually the first demonstration of an important ingredient for spin based quantum computing, such as the first demonstrations of single-spin measurement (Elzerman et al, Nature), two-spin measurement (Petta et al Science), single-spin electric dipole spin resonance (Nowak et al, Science), single-charge-photon coupling and single-spin-photon coupling (Mi et al, Science and Nature), and the more recent demonstrations of high-fidelity single- and two-spin gates that push the spin qubit systems to the threshold of demonstrating quantum error corrections. For the current manuscript, there are two important results, though I believe only the first, demonstrating exchange gate for two electrons on conveyor belts, could be on par with the progresses mentioned above. The demonstration of quantum teleportation, while by itself quite interesting, is unfortunately only partial, and the level of fidelity is still quite low such that it can only be considered a demonstration of principle.

Thus to me the main question is whether the experimental demonstration of the two-qubit coupling is truly a breakthrough that is different from the existing two-qubit gates well known to the authors and multiple competing groups. Unfortunately I do not feel that I have acquired enough information from the manuscript to make this judgment call. On first sight, it seems that the authors used two conveyor belt pulse sequences to push two spins together. When the two are sufficiently close to each other, the barrier between the electrons vanishes and the two dots merge into a single elongated dot if one continues to carry on with the conveyor belt pulses. Based on this simple physical picture, I cannot see how this exchange gate is different in principle from, say if the authors perform a bucket brigade protocol to move the electrons from dots 2 and 5 to 3 and 4, respectively, and then lower the tunnel barrier between dots 3 and 4, especially considering that usually the conveyor belt potential minima move quite slowly (the speed information is not given in this manuscript), in the same order as when a detuning pulse is applied to a quantum dot. Furthermore, the descriptions of the protocol lack details in terms of the precise timing of different segments of the pulse sequence, making it more difficult to differentiate the current protocol from conventional ones. In short, based on the information presented in the current manuscript, it is difficult for me to see why the two-qubit gate is different from those already widely studied, and why this work deserves to be published in Nature.

I also have a couple technical comments.

First, the description of Fig.1e given in the last paragraph of Section II.A seems to be inconsistent with the actual figure, as if the figure has been changed after the description was written. Please check and correct the paragraph.

Second, the discussions in the paragraphs describing Figs. 2d and 2f seem to have some inconsistencies, particularly with respect to where the optimal operating point is. The earlier paragraph indicates that 0.9 conveyor cycle is a sweet spot balancing stronger exchange coupling and slower decoherence, while the later paragraph points out that after 0.86 conveyor cycle the exchange coupling J saturates, while decoherence increases rapidly BEFORE J saturates. The authors should double check these descriptions to make sure that they are consistent.

I am somewhat confused by the use of the terminology "conveyor cycle" (CC). Such a term carries the notion that one is concerned with multiple cycles, which is certainly true when one uses the conveyor belt mode to carry an electron over a longer distance. However, in the current work the electrons are only moved by up to one unit of interdot distance, such that all the figures showing CC dependences have only fractions of a CC, making these figures less transparent compared to if one simply uses moving distance for the axis and nanometer as the unit.

In summary, while the experimental results reported in this manuscript are impressive, I am not convinced that the paper warrants publication in Nature in terms of the level of impact and importance in the field.

Version 1:

Reviewer comments:

Referee #1

(Remarks to the Author)

Referee #2

(Remarks to the Author)

I appreciate the authors' efforts to address the points raised by the reviewers in the previous round. The addition of Supplementary Section B (The concept and advantages of a mobile spin qubit architecture) specifically aims to resolve my main previous concern, and I now feel more confident in recommending this work for publication. However, before I can recommend the manuscript as it stands, I would like the authors to consider the following points related to this section.

1) The authors estimated the number of control electrodes per unit cell for the proposed architectures but did not give the estimated required number of phase-shifted sinusoidal signals for the entire device. I asked for the latter in the previous round because it is a key aspect of the conveyor mode shuttling that impacts scalability [31]. Will signals be independently

controlled or shared (in ideal cases) for different conveyor belts in the proposed architectures?

2) In the main text, two qubit gates are demonstrated only with the serial arrangement of conveyor belts (although it does encourage exploration of other arrangements). Am I correct in understanding that the conveyor directions need to be reversed whenever a spin reaches the interaction zone in the serial configuration? Does this impact the number of qubits that can be simultaneously transported in a single conveyor and be coupled in the interaction zone? If so, I suggest clearly stating these limitations in the text and revising the arrows shown in Fig. 11.

3) I am a bit confused about the number of tones. Does this section assume single-tone shuttling, whereas throughout the main text the authors use two-tone pulses? If that is the case, providing a brief explanation at the beginning of this section would benefit the readers.

Referee #3

(Remarks to the Author)

I have read the revised manuscript as well as the authors' reply to my main concerns. I'm pleasantly surprised that the authors managed to answer my questions and largely removed my doubt about whether the progress they reported is up to par for Nature. I agree with the authors that a two-qubit gate without the need to tune the tunnel barrier specifically would be a significant advance in the field, as such I am convinced now that this work should be published in Nature. The only request I would like to raise to the authors is to include in the abstract and/or the main text explicitly that the protocol they have implemented is done without actively controlling the tunnel coupling between the two dots. This is obviously already implied in the text, though as I read the revisions in the manuscript I did not see it mentioned explicitly. Considering that I myself missed it the first time I read the manuscript, I suspect many others may miss this crucially important point without getting an explicit reminder.

Referee #1 (Remarks to the Author):

Review of the manuscript “Two-qubit logic and teleportation with mobile spin qubits in silicon” by Y. Matsumoto et al., submitted to Nature.

In this work, the authors first demonstrate and benchmark a controlled-phase (CPHASE) gate between two spin qubits in Si/SiGe, and in the second part of the manuscript, they present conditional post-selected quantum state teleportation.

The CPHASE gate between two spin qubits, each encoded in a single electron spin, has been demonstrated previously (e.g., Ref. 13). Conditional teleportation of quantum-dot spin states in Si/SiGe has also been reported [<https://doi.org/10.1038/s41467-020-16745-0>]. What is new here is that the two electrons—each representing a spin qubit—are mobile and are first shuttled toward each other using conveyor-belt-style shuttling, as described in Refs. [28–30]. Conveyor-mode shuttling is an attractive approach for qubit connectivity, as it appears to require a low number of control signals and can reduce cross-talk.

The combination of such shuttling with a reliable two-qubit gate has not been demonstrated before, and this represents a significant step toward more scalable spin qubit architectures, reconfigurable qubit arrays, and more flexible quantum error correction schemes. It is worth noting that combining already demonstrated spin-qubit functionalities—such as the CPHASE gate and shuttling—while maintaining high fidelity is a non-trivial task, and the authors have achieved this very effectively.

In the second part of the manuscript, the authors go one step further by integrating the shuttling and CPHASE gate into a conditional quantum teleportation protocol over a distance of four quantum dots. The protocol is primarily limited by unoptimized detection fidelity.

Beside the impressive experimental demonstration, the manuscript stands out by benchmarking fidelities and data analysis and transparent presentation of the data. Many details are given in the methods and supplementary information. It is also highly appreciated that the authors explore the regime of strong electron-electron interaction and saturation of exchange interaction, although this analysis reveals some unfortunate region.

This paper is a highlight on spin-qubit quantum computing. A publication in Nature is definitely recommended.

I would like to invite the authors to address the following items:

We thank the referee for reviewing our manuscript. We appreciate the positive comments and the recommendation for publication in Nature. We also thank the referee for their comprehensive analysis of our work and the numerous and insightful questions, to which we respond point by point below.

Most important:

1. Fine adjustments of gates: How are the DC offsets and phase offsets adjusted (Tables I to III)? How important is this adjustment? If the adjustment is significant, this should be discussed in the conclusion. It seems that these adjustments are still a complication which is worth to be solved in the future.

The DC offsets during conveyor shuttling have two main reasons: First, we find that strongly isolating double quantum dots 1-2 and 5-6 during readout provides the best readout quality and stability. During conveyor operation, the dot that starts close to below P2 is gently moved to roughly below P3 (and similar for the other side). The readout configuration and conveyor configuration require different voltages on several gates. To transition from the readout to the conveyor configuration, we pulse the offsets on several gates as shown in the tables. In addition, for the high-fidelity operation of the two-qubit gate, the voltages on plunger gates P3 and P4 must be such that the potential is not too far from charge symmetry between Q2 and Q5 (Table II).

The phase offset on each gate is simply set to create the traveling wave potentials, where only the global initial phase of the conveyors was chosen such as not to interfere with the readout charge occupation.

The reviewer correctly points out that individual tuning of the DC offsets could well pose complications for future large processors. The DC offsets are in principle mainly of importance either close to a readout or interaction region. We foresee that there will be no need for individually tuned DC offsets along extended conveyor channels, as potential disorder along the channel can be compensated by increasing the conveyor amplitude [30], which improves the robustness to such variations.

We have added a comment on this matter in the conclusion, as proposed by the reviewer.

For clarification of their meaning, we opted to change 'DC offsets' to 'pulsed offsets' in the manuscript. The offsets are pulsed by the AWG though they are present throughout the entire conveyor operation.

2. The "unfortunate" low T_2^* region is very interesting. Do I understand correct that the regime in Fig. 2e should be better in theory, but cannot be exploited due to this unfortunate region? Is the elongated QD more exposed to dephasing? What is known about this unfortunate region? Do valley, orbital excitations play a role? How can such a region be avoided?

The reviewer is correct in stating that the theoretically attractive regime of Fig. 2e cannot be exploited due to the unfortunate low T_2^* region. The region itself cannot be avoided because it is a fundamental consequence of the process used to merge the two potential wells. While merging, the exchange coupling (J) is exponentially sensitive to the controlling barrier voltage. Therefore, any finite time spent ramping the voltage to traverse this highly sensitive region will cause dephasing, which becomes significantly worse at slower transition speeds.

However, this dephasing effect can be mitigated. One approach to minimizing dephasing is to engineer a lower saturation value of J , as elaborated in response to the next question. Another is to minimize the transit time through this region by increasing the ramp speed (i.e., the effective

bandwidth) of the Arbitrary Waveform Generator (AWG). This faster shuttling would not only suppress dephasing but also transform the interaction from a CPHASE into a SWAP-style gate.

3. By comparing the two CPHASE regimes in Fig2, is there an optimal or just better gate pitch to further exploit the strong electron-electron regime? Or can the CPHASE gate be performed with any reasonable gate-pitch?

By adjusting the gate pitch or optimizing the potential merging process, it may be possible to reach the merged potential regime with a lower saturation value of J , which would reduce dephasing during the ramping in. This could enable higher-fidelity gates. We have added supplemental information detailing simulations on how the tunnel coupling between potential minima belonging to the two conveyor potentials can be controlled via potential shape and center barrier voltages.

In these simulations, the tunnel coupling saturates at the orbital energy splitting of the merged dot. In reality, J saturates at a lower value due to electron-electron interactions, an effect influenced by the anisotropy of the electrostatic potential's shape. Simulating this more accurately would require exact diagonalization or even full configuration interaction simulations, which is computationally expensive and was not performed.

We conclude that a lower saturation value for J , requiring a more elongated potential, will limit dephasing during ramp-in due to suppressed sensitivity to high-frequency noise in the exchange coupling. However, in the merged regime in Fig. 2(f) where J saturates, we believe that T_2^* will eventually be limited by the increased electric dipole of the elongated potential. The corresponding increased electrical susceptibility enhances single-spin dephasing in the magnetic field gradient. This trade-off likely leads to an optimal elongation of the dots with a corresponding gate pitch in the interaction zone of the device.

Lastly, for an adiabatic CPHASE gate, a large Zeeman energy difference relative to the saturated J is preferable. However, this aggravates single-spin dephasing from the Zeeman gradient.

Consequently, we believe that diabatic gates like the \sqrt{SWAP} , which can be performed even with a small Zeeman gradient, may ultimately work even better.

We have added a discussion of these trade-offs in supplementary section D and refer to it in Results B in the main manuscript.

4. Please specify the shuttle distances and the quantum teleportation distance in the abstract.

We have added the shuttle distance and the quantum state teleportation distance in the abstract.

Minor:

5. Does the shuttling (125 MHz with high amplitude) contribute to the heating? Would it contribute if shuttling is across longer distances?

We acquired data regarding the heating effect of shuttling on the qubits, albeit at a much lower external magnetic field than used in this work. We do not really observe a clear qubit frequency shift with increased back-and-forth conveyor shuttling, however, the mixing chamber temperature increases from about 20 to 40 mK, which is aggravated with longer shuttling sequences. We

cautiously conclude that shuttling in the 10-300 MHz range does indeed contribute to heating, although far less than driving single-qubit rotations using microwave control above a few GHz. The reviewer astutely points out the need to further investigate power dissipation when scaling the length of a conveyor channel. Since our conclusions are only tentative at the moment, we prefer not to comment on them in the present manuscript.

6. If you have information about the valley splitting of the device, please add it to the methods/supplements.

We have added the table of the valley splittings measured on this device in supplementary Table I .

Referee #2 (Remarks to the Author):

The work by Y. Matsumoto reports on two-qubit logic for silicon spin qubits in a unique scheme. The experiments are elaborated and the results are enlightening. The authors furthermore claim its substantial implication for future silicon-based quantum processors, which is of high relevance to a broad readership. The manuscript is certainly publishable in a high-profile journal, but my view is that its impact - whether it presents the level of an advance expected for a Nature paper - hinges critically on the benefits of the proposed reconfigurable architecture for large-scale integration based on the demonstrated two-qubit gate operation.

We thank the referee for reviewing our manuscript and we appreciate their insightful questions. We address the referee's questions and concerns below.

The main point:

1) I find it hard to see from the manuscript how the authors propose to implement the mobile-spin two-qubit logic with finite sets of independent conveyor potentials in a large-scale system. What is the estimated required number of phase-shifted sinusoidal signals (not the tones) in the entire qubit array, in principle, for a reconfigurable, flexibly connected 2-dimensional qubit array of $n \times m$ - where n is the number of conveyor rows and m is the number of interaction zones along a conveyor row? Can the authors illustrate the idea e.g. in Fig. 1a?

We thank the referee for this crucial question regarding the scalability and practical implementation of our architecture. This point is central to the impact of our work. To clarify our proposal for implementing quantum logic in a large-scale, reconfigurable system, we have incorporated a detailed discussion on potential architectures in the supplementary information, which we summarize here.

First, the core concept of the mobile spin qubit operation is to perform two-qubit gates using two independent conveyor channels and a single central barrier gate. Our scheme can be extended to large-scale $n \times m$ arrays using two primary approaches, each offering a different trade-off between control complexity and qubit connectivity.

1. Serial "Shelf" Architecture (Fig. 10(a)): Full Connectivity

As an example of a mobile spin qubit architecture that maximizes connectivity, we propose the serial “shelf” architecture illustrated in Fig. 10(a). Here qubits are stored in the storage zones above and below the conveyor, and are transported to an interaction zone inside the conveyor for implementing two-qubit gates.

- **Implementation and scaling:** This architecture arranges qubit interaction/storage zones serially along the conveyor belts. While the number of sinusoidal signals for the two conveyors remains constant (8), this design requires two additional static gates for each storage zone. The number of storage zones m is proportional to the number of qubits N . For a single conveyor row ($n=1$) with $m = N$ storage zones, the total number of control lines would scale as 8 (two conveyors) + 1 (barrier) + $2 * N_{\text{qubit}}$ (storage zones).

- **Connectivity:** The primary benefit of this approach is its ability to achieve all-to-all connectivity among all qubits within that row, as shown in Fig. 10(e). This is accomplished by dynamically operating the central barrier gate to allow shuttling of a qubit from one side to the other, allowing two-qubit gates between any pair of qubits in the system.

- **Scaling in a plane:** An example of how such unit cells can be arranged in a two-dimensional plane is shown in Fig. 11(a). The total number of storage zones is still equal to the number of qubits N but the storage zones that are placed in between two horizontal conveyor channels now require three instead of two gates each. The total number of conveyor channels n is given by N/m with m the number of storage zones per conveyor. The diagram in Fig. 11(a) also shows an implementation with k operation zones per conveyor, which allows more operations to be performed in parallel at the cost of additional control lines. The total number of control lines is then given by $n * (4*(k+1)) + 2m$.

2. Parallel Architecture: Constant Control Overhead

A variant of the mobile-spin-qubit scheme which we call the parallel architecture is shown in Fig. 10(b). Here the qubits are stored in the conveyor, and interaction zones are placed between the conveyors. This design also serves as the fundamental repeating unit of a larger array.

- **Implementation and scaling:** In this configuration, two conveyor belts run in parallel, separated by a single barrier gate. The crucial advantage here is that the number of required control signals (the phase-shifted sinusoids) is constant and independent of the number of qubits. For each pair of conveyors, a fixed set of $4 * 2 + 1 = 9$ electrodes is sufficient to control an arbitrary number of qubits N , which are here stored in two conveyors. This represents a significant reduction in control overhead compared to conventional architectures. The practical limit is determined by the shuttling fidelity over long distances, not the number of control lines.

- **Connectivity:** This design provides all-to-all connectivity between the two adjacent conveyor rows, as depicted in the connectivity graph in Fig. 10(f). This is ideal for many quantum algorithms, particularly quantum error-correction codes, where data and ancilla qubits can be placed on separate conveyors, requiring frequent interaction between the two sets but not necessarily within them [54].

- Scaling in a plane: A representative two-dimensional layout of this unit cell is illustrated in Fig. 11(b). The number of barrier gates (interaction zones) k between adjacent conveyors is a key design parameter, determined by the limits of shuttling fidelity and the connectivity required for achieving fault-tolerance. The number of qubits N is now distributed over n conveyors, which each host $m = N/n$ qubits. This amounts to a total number of control lines given by $k + 4*n$.

3. Considerations on the number and spacing of interaction zones k

For both the serial and parallel architectures, the maximum number of qubits N that can share a given interaction zone is determined by the condition

$$\epsilon_{2Q} + \epsilon_{\text{shuttle}} < \epsilon_{\text{FT}} (\sim 0.8\%),$$

where ϵ_{2Q} is the two-qubit gate error and $\epsilon_{\text{shuttle}}$ is the cumulative error from moving qubits to and from the interaction zone.

As reported in [30], an effective 10 μm shuttling distance can be achieved with 99.5 % fidelity. Based on these numbers, if storage zones are placed every 250 nm along a conveyor row, up to 20 storage zones can be accommodated adjacent to the top and bottom rows of a conveyor. Figures 10 and 11 illustrate layout examples in which all storage zones are populated with qubits. Alternatively, as in Fig. 1(a), empty storage zones can be deliberately left to allow dynamic shuffling of qubit positions depending on the connectivity or algorithmic requirements.

In the parallel architecture, the same condition applies to the longest possible shuttle path—for example, bringing the left-most qubit in the upper row to interact with the right-most qubit in the lower row.

In addition, the idling fidelity of qubits in storage zones imposes a further constraint:

$$N_{\text{seq}} * (t_{\text{shuttle}} + t_{2Q}) \ll T_2^{\text{storage}},$$

where N_{seq} is the number of sequential two-qubit gates per algorithmic cycle, t_{shuttle} is the shuttle time per hop, and t_{2Q} is the two-qubit gate time. Characterizing T_2^{storage} at magnetically quiet storage zones without micro-magnets will be crucial for estimating this limit. Our current devices achieve $t_{2Q} \sim 50$ ns for CZ gates, with $t_{\text{shuttle}} < 200$ ns over effective 10 μm [30].

The two architectures presented here are illustrative examples that demonstrate how the scheme developed in this work can substantially improve reconfigurability in large-scale spin-qubit systems. The quantitative error-budget considerations above show that, with the reported shuttling fidelities and gate times, both example designs are compatible with fault-tolerant thresholds, which requires T_2 in the storage zones is sufficiently long. We hope that these examples will serve to illustrate the promise of our results and highlight architectural elements that can be incorporated in future quantum processors. In fact, one can envision many other variants that would scale favourably, building on the same core mechanism. Of particular interest would be sequences designed to facilitate transversal operations between logical qubits, similar to those

demonstrated recently using Rydberg atoms, which we think should also be feasible with spins in conveyors.

Other points:

2) How are the potentials shown in Figures 2c and e simulated? The simulated potential seems to be always symmetric - does the so-called symmetric operation come for free in this two-qubit gate scheme?

The simulations in Figures 2c and e are indeed symmetric because they represent an idealized case without the potential disorder present in the physical device. In the current experiment, a major source of this disorder stems from the specific device layout: the gates at the ends of the array must be biased asymmetrically to optimize the charge sensors and readout fidelity. These biases distort the potential landscape and propagate towards the interaction zone, making fine-tuning for symmetry in that region necessary.

This situation is closely related to the pulsed DC-offset adjustments discussed in response to Referee 1. In both cases, localized tuning of the readout zones is needed to compensate for non-uniformities caused by the current small-scale device architecture. In a larger, scaled-up processor, interaction zones could be placed far from the readout zone, and fabrication improvements would yield a much more uniform potential.

We have now included a detailed description of the simulation methodology in the supplementary information, along with an explicit connection to the discussion of DC-offset adjustments.

3) In the present study, the two-qubit fidelity is about 99%. What would be key in further improving the fidelity in this scheme?

As stated in the main text, we estimate that dephasing due to low-frequency noise contributes $\sim 0.22\%$ to the infidelity of the two-qubit gate. We suspect that the main contribution to the infidelity is due to uncontrolled single-qubit frequency shifts from heating effects [24], which also impact the two-qubit gate. Therefore, the key to further improve fidelity in this scheme is to reduce heating effects. One route is to reduce on-chip dissipation—most directly by lowering the series resistance of the gate electrodes during fabrication, for example by reducing gate/electrode sheet resistance and contact resistance through thicker metal layers, lower- ρ stacks, or larger vias, or using superconducting wires and contacts to largely eliminate Ohmic losses. The other route is to reduce the sensitivity of the qubit frequencies to temperature. In our lab, we notice device-to-device variability of this sensitivity and some other groups do not report noticeable heating effects. A better understanding of the underlying physics should lead to qubit frequencies that are (largely) insensitive to temperature.

Once heating effects are mitigated, further improvements would require suppressing low- and high-frequency charge noise. Over the years, the Scappucci group has realized systematic reductions in charge noise by optimizing the heterostructure [B. Paquelet Wuetz et.al., *Nature Comm* volume **14**, 1385 (2023)]. It can be expected that a similar systematic effort focused on the surface and the gate dielectrics, and possibly better filtering of the control signals, will bring the

fidelity closer to the ultimate limit set by the qubit coherence during the CZ.

4) Can the authors explain the expected key difference of the presented architecture (Fig. 1a) in terms of the electronics and wiring resources for high-fidelity two-qubit gates from the other existing schemes, where the two-qubit logic that demands for most precise tuning are handled by dedicated gate electrodes?

Following on from the discussion of scalability, architectures based on two-qubit gates acting on mobile spins also offers significant advantages in terms of the electronics and wiring resources required for high-fidelity gates, differing from conventional schemes in several ways. As detailed in our response to the main point, the “parallel” architecture offers a *constant control overhead*, requiring only 9 RF lines per pair of conveyor rows, regardless of the number of qubits. In the “shelf” architecture, only the gates in the interaction zones must be carefully tuned for high-fidelity two-qubit logic (loading spins into the conveyors does not demand the same level of fine-tuning as two-qubit gates). This is in stark contrast to conventional schemes with static qubits, which require dedicated, precisely tuned gate electrodes and control lines between any pair of neighbouring qubits. Furthermore, because the storage zones are spatially separated from the interaction zones and from each other, spectator qubits are well isolated from the local gate pulses in the interaction zones, reducing electrical cross-talk.

5) What will be the impact of the pulse sequence of a CZ gate (shown in Fig. 3b) to other qubits that are being transported in conveyor channels?

We thank the reviewer for raising this point. In the scenario where additional qubits are present in the same conveyor channel, the CZ-gate operation between two target qubits does not induce any special interaction or disturbance to the other qubits. Apart from the pulses applied to the individual center-barrier gate used for the CZ operation, the other qubits in the channel experience only the standard conveyor-mode shuttling pulses. As a result, their evolution is identical to that during regular transport, namely the accumulation of dynamical phase due to position-dependent Zeeman energy along the conveyor path. No additional coupling or phase shift is introduced by the CZ-gate pulses themselves. We clarify this point in the caption of Supplementary Figure 10.

6) The authors conclude in the supplementary section that somehow the teleportation process fidelity depends rather strongly on the Bell measurement outcome. In my opinion, this should be mentioned in the main text.

We agree and added the statement that the teleportation process depends on the Bell measurement outcome in the main text.

Minor comments and suggestions.

7) For improved readability, please specify somewhere that the qubit 0 state corresponds to the spin-down state in this work.

We now explicitly define that the qubit 0 state is spin-down in the discussion of Fig. 1e.

8) In the caption of Fig. 1(c), the color assignment seems swapped.

We thank the reviewer for pointing this out. Indeed the color assignments for the barrier and plunger gates in Fig. 1(c) were inadvertently swapped. We have corrected both the figure and the caption in the revised manuscript.

9) I suspect that the authors meant a " $Ry(-\pi/2)$ gate" by a " $-Ry(\pi/2)$ gate" (used in Sec. II D).

We appreciate the careful reading. This was a typographical error: we intended $Ry(-\pi/2)$. The text has been corrected throughout Methods D.

10) I suspect that the section title of Methods sec. B contains a typo ("tone" instead of "tune").

Thank you for catching this. We have corrected the section title to "tone."

11) In supplementary Tables I - III, some phase offsets exceed 2π - can the authors double check?

We intentionally reported unwrapped phases (which may exceed 2π) to make the moving direction explicit. To avoid confusion, we have added a note in the captions explaining this convention.

12) Supplementary Table III is not referred to in the main text when appropriate. In addition, it would be helpful to specify for which dataset the configuration is used in the table caption.

We now cite Supplementary Table III at the appropriate locations in the main text, and the table caption specifies the dataset to which the configuration applies.

13) Can the authors double check the description of the error bars in the caption of Fig. 7?

We double-checked the definition of the error bars. They are correct.

14) In Fig. 8, is the AC voltage offset the same as the DC offset?

The two terms refer to the same quantity. We have harmonized the terminology and made it consistent across the figure and caption.

15) In Fig. 8, what do the error bars mean in the last sentence of the caption? Should it be deleted?

This was a leftover typographical error; we have removed it.

16) In Supp. Sec. E, the qubit frequencies are wrongly labeled, if I am not mistaken.

We thank the reviewer for the careful reading. The labels were indeed incorrect and have been corrected in the revised supplement.

We hope these revisions address the reviewer's questions on the benefits of the proposed reconfigurable architecture for large-scale integration based on the demonstrated two-qubit gate operation. In addition to these important benefits, the two-qubit gate implemented directly on

mobile spins allows in principle to perform different types of two-qubit gates simply by varying the conveyor speed. We have already obtained initial results in this direction, which we elaborate on in the response to Referee #3. This possibility further amplifies the significance of the present result.

Referee #3 (Remarks to the Author):

In the manuscript titled "Two-qubit logic and teleportation with mobile spin qubits in silicon", the authors implemented a protocol in which they performed transportation and coupling of two electron spins through a single pulse sequence. After demonstrating the effectiveness of this protocol, the authors performed a partial demonstration of quantum teleportation in such a system. I find the results of both sets of experiments quite impressive, with a controlled-Z gate fidelity of nearly 99% and the teleportation fidelity of 87%. While I have no doubt that the authors have made significant progress in their ability to control electron spins in Si, I am still ambivalent on whether the advances reported in this manuscript is worthy of a Nature article, as I discuss below.

My criterion for the importance/impact of an article is based on what are represented by past Nature and Science articles in the field of spin qubits in semiconductors. What I found from those articles are usually the first demonstration of an important ingredient for spin based quantum computing, such as the first demonstrations of single-spin measurement (Elzerman et al, Nature), two-spin measurement (Petta et al Science), single-spin electric dipole spin resonance (Nowak et al, Science), single-charge-photon coupling and single-spin-photon coupling (Mi et al, Science and Nature), and the more recent demonstrations of high-fidelity single- and two-spin gates that push the spin qubit systems to the threshold of demonstrating quantum error corrections. For the current manuscript, there are two important results, though I believe only the first, demonstrating exchange gate for two electrons on conveyor belts, could be on par with the progresses mentioned above. The demonstration of quantum teleportation, while by itself quite interesting, is unfortunately only partial, and the level of fidelity is still quite low such that it can only be considered a demonstration of principle.

Thus to me the main question is whether the experimental demonstration of the two-qubit coupling is truly a breakthrough that is different from the existing two-qubit gates well known to the authors and multiple competing groups. Unfortunately I do not feel that I have acquired enough information from the manuscript to make this judgment call. On first sight, it seems that the authors used two conveyor belt pulse sequences to push two spins together. When the two are sufficiently close to each other, the barrier between the electrons vanishes and the two dots merge into a single elongated dot if one continues to carry on with the conveyor belt pulses. Based on this simple physical picture, I cannot see how this exchange gate is different in principle from, say if the authors perform a bucket brigade protocol to move the electrons from dots 2 and 5 to 3 and 4, respectively, and then lower the tunnel barrier between dots 3 and 4, especially considering that usually the conveyor belt potential minima move quite slowly (the speed information is not given in this manuscript), in the same order as when a detuning pulse is applied to a quantum dot. Furthermore, the descriptions of the protocol lack details in terms of the precise timing of different segments of the pulse sequence, making it more difficult to differentiate the current protocol from conventional ones. In short, based on the information presented in the current manuscript, it is difficult for me to see why the two-qubit gate is different from those already widely studied, and why this work deserves to be published in Nature.

We sincerely thank the referee for reviewing our manuscript and for their positive comments regarding the experimental progress we have made in controlling electron spins. We appreciate and agree with the referee's perspective on the publication criteria for Nature and Science, which typically report the first demonstration of a key building block in semiconductor quantum computing. We are convinced the present work falls in this category.

We believe that this first demonstration of a two-qubit gate of spins on the move will spark a broad shift towards semiconductor spin qubit architectures with increased qubit connectivity. Indeed, recent advances with neutral atom qubits have demonstrated the power of reconfigurable two-qubit interactions.

The referee brings up an important question on how the conveyor protocol differs from a bucket brigade implementation to shuttle the two electrons to static dots on which the two-qubit gate would be executed in the conventional way. To be clear, this approach would also allow reconfigurable two-qubit interactions. However, the essential differences with the conveyor implementation can be seen from the important benefits of the conveyor-based gate which acts directly on two mobile spins:

1. Compared to bucket-brigade shuttling and exchange control via the interdot barrier of a predefined double quantum dot, the conveyor approach reduces the number of individual control lines, easing scalability requirements. To realize this experiment with bucket-brigade shuttling, one would need to independently and precisely control the tunnel coupling across the entire array. In contrast, the conveyor implementation requires just four control signals regardless of the distance over which the electrons are shuttled [Xue, De Smet/Matsumuto].
The conveyor implementation is very fast, different from what the referee anticipated. In [De Smet/Matsumuto], we report high-fidelity conveyor shuttling at a speed over 50 m/s, covering an effective distance of 10 micron in under 200 ns. This speed is hard to match in a bucket-brigade implementation. The current practical limitation on the maximum speed arises from the bandwidth of the AWG, which requires pulse durations of a few nanoseconds in both the bucket-brigade and conveyor styles. This limited the speed to 22.5 m/s for the segments where the electrons move toward each other before they interact. In addition, the motion must remain adiabatic in the orbital and valley sector, and for a CZ operation also remains adiabatic in the spin sector (diabatic behavior in the spin sector can in fact be leveraged in a highly original way, as discussed under point 6 below). Therefore, we reduce the speed to 1.8 m/s when activating the exchange interaction.
2. The current practical limitation on the maximum speed arises from the bandwidth of the AWG, which requires pulse durations of a few nanoseconds in both the bucket-brigade and conveyor styles in our case.
3. The conveyor shuttling fidelity is much higher than the bucket-brigade shuttling fidelity over the same distance. In [De Smet/Matsumuto], we reported that the phase flip probability is 10 times lower with conveyor than with bucket-brigade shuttling. The reason is that in bucket-brigade shuttling, the electron passes through the avoided crossing of successive pairs of dots, where dephasing is enhanced.

4. In a hybrid implementation where the electrons are moved towards static dots by a conveyor, the transition from the conveyor channel to the static dot introduces errors which can be avoided when acting on the electrons while they remain in the conveyor dots. We note that similarly, errors occur when loading an electron from static dots 2 and 5 into the conveyor and back as part of initialization and readout. The reported two-qubit gate fidelity of 99% is therefore a lower bound, as it includes those loading and unloading errors.
5. The conveyor-based gate allows further optimization of the gate operation by selecting the precise position where the two-qubit gate is implemented (not yet demonstrated here). In function of the magnetic field gradients produced by the micromagnet, we can move spins into regions of higher coherence and precisely select the Zeeman splitting difference between them by choosing the interaction position, in order to boost the gate fidelity.
6. Lastly, and this is perhaps the most fundamental point, the two-qubit gate controlled by conveyor potentials can be used to implement *in a single step* multiple types of two-qubit gates, such as SWAP, CZ and CX gates, *simply by adjusting the conveyor speed*. This requires tailored but achievable magnetic field gradients, as we demonstrate theoretically in very recent work (<https://arxiv.org/abs/2508.08394>). We already obtained preliminary data supporting a similar idea. **Redacted**

Redacted

Through the combination of the architectural advantages of conveyor-based two-qubit gates and the capabilities of velocity-based gate control, the power of acting directly on mobile spins, which is the key novelty demonstrated here, becomes even more apparent.

We have summarized the points above in the discussion section of the manuscript, leaving the data shown above for a future publication. In Section C, we have added details on the estimated velocity to the description of the pulse shaping protocol for Figure 3b.

I also have a couple technical comments.

First, the description of Fig. 1e given in the last paragraph of Section II.A seems to be inconsistent with the actual figure, as if the figure has been changed after the description was written. Please check and correct the paragraph.

We value this comment from the reviewer and recognize the confusion that the description of Fig 1e in Section II.A brings. While this figure has not been changed after writing its description, the use of ‘panels’ – referring to the two sides created by the vertical dashed line at 11.9 GHz – might be misleading here. We have rewritten the text such that the description conveys more clearly the different spin initializations for both sides of the figure.

Second, the discussions in the paragraphs describing Figs. 2d and 2f seem to have some inconsistencies, particularly with respect to where the optimal operating point is. The earlier paragraph indicates that 0.9 conveyor cycle is a sweet spot balancing stronger exchange coupling and slower decoherence, while the later paragraph points out that after 0.86 conveyor cycle the exchange coupling J saturates, while decoherence increases rapidly BEFORE J saturates. The authors should double check these descriptions to make sure that they are consistent.

The reviewer correctly notices the different conveyor cycle values for Figs. 2d and 2f. We stress that, different from Fig. 2d, the configuration for Fig. 2f is at an optimal operating point. The mentioned conveyor cycle value of 0.86 only indicates the saturation of the exchange. Decoherence indeed increases rapidly before this point, as this is where the exchange coupling is very sensitive to electrical noise.

I am somewhat confused by the use of the terminology "conveyor cycle" (CC). Such a term carries the notion that one is concerned with multiple cycles, which is certainly true when one uses the conveyor belt mode to carry an electron over a longer distance. However, in the current work the electrons are only moved by up to one unit of interdot distance, such that all the figures showing CC dependences have only fractions of a CC, making these figures less transparent compared to if one simply uses moving distance for the axis and nanometer as the unit.

We understand that the term 'conveyor cycle' is perhaps not an immediately intuitive way to describe the control of the position of the electrons and we have discussed extensively possible alternative formulations, both before submitting and again triggered by the referee's question. Our choice to use this term stems from the fact that, due to small variations in the potential landscape, the electron motion is not necessarily perfectly linear and true positions (in nm) cannot be accurately determined. Therefore, we believe that reporting on the control of the motion (in conveyor cycles) is the most reliable approach. Since the distance does remain more intuitive, we had already provided the nominal displacement in nm in Fig. 1 and Fig. 2. These were labelled "potential displacement (nm)" in the original submission and we added a clarification that the nominal displacement may slightly differ from the actual displacement of the potential minima.

In summary, while the experimental results reported in this manuscript are impressive, I am not convinced that the paper warrants publication in Nature in terms of the level of impact and importance in the field.

We hope that our response above has convinced the referee of the conceptual novelty of the two-qubit gate implementation by conveyor potentials, and the important advantages and impact offered by operating on spins in conveyor potentials versus conventional methods based on bucket-brigade shuttling and a two-qubit gate acting on electrons in static dots, making Nature the appropriate venue for this work.

Referees' comments:

Referee #2 (Remarks to the Author):

I appreciate the authors' efforts to address the points raised by the reviewers in the previous round. The addition of Supplementary Section B (The concept and advantages of a mobile spin qubit architecture) specifically aims to resolve my main previous concern, and I now feel more confident in recommending this work for publication. However, before I can recommend the manuscript as it stands, I would like the authors to consider the following points related to this section.

1) The authors estimated the number of control electrodes per unit cell for the proposed architectures but did not give the estimated required number of phase-shifted sinusoidal signals for the entire device. I asked for the latter in the previous round because it is a key aspect of the conveyor mode shuttling that impacts scalability [31]. Will signals be independently controlled or shared (in ideal cases) for different conveyor belts in the proposed architectures?

The number of control electrodes per unit cell provides an estimate of the local hardware complexity. However, executing an arbitrary quantum circuit on physical qubits indeed requires a corresponding number of phase-shifted sinusoidal signals distributed across the device, which is a central aspect of conveyor-mode shuttling and directly impacts scalability.

In the most general case, each unit cell would require an independent set of phase-shifted signals to enable fully flexible shuttling operations. This would imply that the required number of DAC channels scales with the number of unit cells. Nevertheless, the effective number of DACs can be substantially reduced by employing specialized cryogenic and room-temperature electronics, such as multi-phase signal generators or delay-line-based phase shifters, as discussed for example in Ohira et al., 2025 arXiv:2508.04093. Such techniques allow one DAC to generate multiple phase-shifted outputs, thereby strengthening the scalability of conveyor-mode shuttling.

In some future scenarios, the signals may not need to be independently controlled for all conveyor belts. Since fault-tolerant quantum computing relies on transversal gate operations, the logical qubit layout and gate scheduling impose strong structure on the required control patterns. By optimizing the spatial assignment of logical qubits and

restricting operations to those needed for transversal gates, possibly several control signals can be shared across multiple conveyor belts in ideal cases. In this regime, global or semi-global phase-shifted signals can be broadcast to multiple unit cells, while only a limited subset of control lines remains locally tunable for calibration and error mitigation.

In summary, while fully general operation would require independently controllable phase-shifted signals per unit cell, practical implementations may significantly reduce the number of required DACs by (i) hardware-level signal generation and sharing, and (ii) architectural and logical-level optimization enabled by transversal gate sets.

2) In the main text, two qubit gates are demonstrated only with the serial arrangement of conveyor belts (although it does encourage exploration of other arrangements). Am I correct in understanding that the conveyor directions need to be reversed whenever a spin reaches the interaction zone in the serial configuration? Does this impact the number of qubits that can be simultaneously transported in a single conveyor and be coupled in the interaction zone? If so, I suggest clearly stating these limitations in the text and revising the arrows shown in Fig. 11.

We appreciate the reviewer's insightful comment regarding the limitations of simultaneous two-qubit interactions in the serial conveyor-belt configuration.

In the serial arrangement shown in the main text, the reviewer's understanding is essentially correct: when a spin reaches the interaction zone, the conveyor direction must be reversed after the two-qubit gate. As a consequence, the number of qubits that can be simultaneously transported on a single conveyor belt and coupled within the same interaction zone is limited. In this sense, the achievable degree of parallelism for two-qubit gates indeed becomes a bottleneck of the purely serial configuration.

One possible mitigation strategy is to implement two-qubit gates not only between conveyor channels, but also between a conveyor channel and a nearby storage zone. These gates could be $\sqrt{\text{SWAP}}$ gates, which require no additional components, or CPhase-type gates, which require an additional nanomagnet in proximity to the storage region. However, introducing nanomagnets comes at the cost of increased magnetic field gradient and enhanced decoherence in the storage zones.

Therefore, the optimal architecture depends on a nontrivial balance between gate parallelization, two-qubit gate fidelity, and coherence when idling and during transport. In the revised version, we have clarified the constraints on simultaneous two-qubit interactions in the serial configuration and revised the arrows in Fig. 11 to better reflect the required conveyor operation during two-qubit gate execution.

3) I am a bit confused about the number of tones. Does this section assume single-tone shuttling, whereas throughout the main text the authors use two-tone pulses? If that is the case, providing a brief explanation at the beginning of this section would benefit the readers.

We thank the reviewer for pointing out the potential confusion regarding the number of tones.

Throughout this work, we primarily employ two-tone conveyor operation. The purpose of using two phase-shifted sinusoidal tones is to ensure that the nearest-neighbor potential minima are not unintentionally hybridized or coupled during shuttling.

The section in question does not assume a fundamentally different shuttling mechanism; rather, it discusses the control requirements in a more general framework. Conceptually, similar isolation of the moving potential minima could also be achieved with a single-tone approach if other design or operational parameters are optimized. For example, this could be realized by (i) optimizing the gate pitch at the device-design level (Langrock et.al., 2023 PRX Quantum), (ii) increasing the amplitude of the sinusoidal drive, or (iii) reducing the static potential disorder in the device. In practice, the two-tone approach offers a convenient and experimentally robust solution without requiring aggressive assumptions on fabrication precision or drive amplitudes.

To avoid ambiguity, we have added a brief explanatory paragraph at the beginning of the corresponding section, explicitly stating that two-tone shuttling is used in the experiments and clarifying under which conditions a single-tone scheme could be sufficient.

Referee #3 (Remarks to the Author):

I have read the revised manuscript as well as the authors' reply to my main concerns. I'm pleasantly surprised that the authors managed to answer my questions and largely removed my doubt about whether the progress they reported is up to par for Nature. I agree with the authors that a two-qubit gate without the need to tune the tunnel barrier specifically would be a significant advance in the field, as such I am convinced now that this work should be published in Nature. The only request I would like to raise to the authors is to include in the abstract and/or the main text explicitly that the protocol they have implemented is done without actively controlling the tunnel coupling between the two dots. This is obviously already implied in the text, though as I read the revisions in the manuscript I did not see it mentioned explicitly. Considering that I myself missed it the first time I read the manuscript, I suspect many others may miss this crucially important point without getting an explicit reminder.

Conceptually, we agree with the reviewer that a key feature of our approach is that the two-qubit gate does in principle not require actively tuning or optimizing the tunnel coupling as an independent control parameter, which indeed distinguishes our protocol from many existing implementations.

However, the literal formulation that the tunnel coupling is “not controlled” is not strictly accurate. The gate B3, which controls the tunnel barrier when the two dots are in their closest position, receives a sinusoidal signal as part of the traveling-wave (conveyor) potential, together with the other gates. In the data shown in Fig. 2f, this sinusoidal signal has the same amplitude as on the other conveyor gates (see Table IV). For the data in Fig. 2d, Fig. 3, and Fig. 4, a slightly different conveyor amplitude is applied to B3 compared to the other gates (see Table III), although it is not strictly necessary. In addition, as detailed in the manuscript, static offsets are applied to all gates to compensate for sample-specific inhomogeneities.

As discussed in the Conclusion, such individual offsets are not required in sufficiently homogeneous samples (see Ref. [29] and arXiv:2510.26860). Therefore, while the reviewer’s interpretation is correct at a conceptual level, stating that the tunnel coupling is not controlled at all in the present experiment would be misleading.

To address this point while maintaining technical accuracy, we have clarified the role of gate B3 and the tunnel coupling explicitly in the discussion of Fig. 2f in the main text. We believe that this allows us to convey the necessary nuance without oversimplification, whereas including a simplified statement in the abstract could lead to misunderstandings.